# Analysis of the Mixing of Filler and Base Materials in Arc-Welded Single-Bead Surface Welds Using an EDXS Method

**DOI:** 10.3390/ma15010217

**Published:** 2021-12-28

**Authors:** Borut Zorc, Matija Zorc, Aleš Nagode

**Affiliations:** 1Faculty of Natural Sciences and Engineering, University of Ljubljana, Aškerčeva 12, 1000 Ljubljana, Slovenia; matija.zorc@ntf.uni-lj.si (M.Z.); ales.nagode@ntf.uni-lj.si (A.N.); 2Welding Institute Ltd., Ptujska 19, 1000 Ljubljana, Slovenia

**Keywords:** covered electrodes, single-bead surface weld, segregation, EDXS analysis, admixing rate

## Abstract

This article deals with an analysis of mixing and determines the admixing rate of a base S355 steel plate in single-bead surface welds by measuring the chemical composition using a plane-scan energy dispersive X-ray spectroscopy (EDXS) on metallographic cross-sections. The results show that obtaining a larger number of EDXS measurements does not necessarily lead to obtaining a more accurate admixing rate. Due to the ever-present segregations that are generally near the base material, the disadvantage of this method is the subjective influence of the SEM operator on the estimated admixing rate. To obtain relevant results, a sufficiently wide area of well-mixed melt, including segregations, must be analyzed. This study showed that by using a sufficiently large number of appropriately selected sites with a sufficiently large surface area, it is possible to estimate the admixing rate from the chemical composition with an accuracy of ≥96% for the geometrically determined admixing rate *D* = 30%. From several equations, the best result showed an equation which is the arithmetic mean of the two different arithmetic means and in which the artificial influencing factor of the segregations of the base material is taken into account. With this equation, the same value of admixing rate, *D* = 30%, was obtained using the comparative geometric method.

## 1. Introduction

During fusion welding, a molten part of the base material is admixed with the melt of the filler material, where the admixing rate of the baes material is about 10–40% [1,2,3] for arc welding with a coated electrode and as much as 70% for submerged arc welding [2]. The weld pool is intensively mixed during arc-welding due to the various operating forces involved. Compared to diffusion in the solid state, the mass transport in the weld pool is much faster due to intensive mixing; therefore, the composition of the resulting weld is fairly uniform over the entire volume. With the known admixing rate of the base material, the average chemical composition of the resulting weld can be calculated [1,3,4]. Several methods for determining the admixing rate of the base material in single-bead surface welds are known. These include using the mass of the melted base and filler materials [5], using the ratio of the area of the metallographic cross-section [1,4,6,7,8,9,10,11], using the ratio of the heights of the metallographic cross-section [1], using the volume fraction of the melted base and filler materials [7,8], using the chemical composition measured by an electron-probe micro-analyzer [1,8], and using the welding parameters and the material thermal-physical quantities [1,7,8]. The influence of the admixing rate on the metallurgical and mechanical properties of multi-weld welds has also been investigated [12,13,14].

Despite the very good mixing of the melt during arc-welding, there are always different types of segregations in the welds. Among these are micro- and macro-segregations, which are the result of a locally poorly mixed welded melt. These are curved with a band-like shape and basically show the mode and motion direction of the melt. These segregations are more pronounced in the welding of two different materials or when using a filler material that is chemically different to the base material [3,6,10,15,16,17,18,19,20]. Due to the good mixing of the melt in arc welding, segregations of this type are primarily limited to weld areas that are closer to the fusion line [3,19,20]. Due to the narrow area of the root part of the weld, these segregations can pass through the entire weld width [10]. Even in welds produced by high-energy processes due to the poor mixing of the melt, this type of segregation is in practice present throughout the weld [6,21]. A feature of fusion welds is also a narrow continuous micro-segregated band of a partially mixed zone near the fusion line, which is conditioned by the different contents of the chemical elements in the filler and the base material. The partially mixed zone is more evident when the difference in the chemical composition of the filler and the base material is larger. The width of this zone is wider when the proportion of the molten base material is greater. For example, in steel welds the partially mixed zone is generally from fifty to a few hundred microns [5,6,7,8,9,15,17,19]. A characteristic of this zone is a very large gradient in the content of alloying elements, which continuously changes from the composition of the resulting weld to the composition of the base material. If the content of an alloying element in the filler material is greater than that in the base material, its content decreases from the weld to the base material in this zone and vice versa.

This article deals with metallographic and EDXS analyses of the mixing of different alloyed-steel filler materials and a base S355 steel in single-bead surface welds that were arc-welded on a S355 steel plate with coated electrodes. This article is a continuation and supplementation of the previous article [1].

## 2. Experimental Procedure

### 2.1. Materials and Preparation of the Samples

Onto a 20 mm-thick non-preheated very low alloy S355J2 steel (with nominal mechanical properties according to EN 10025-2:2019 standard: yield strength *R_e_* ≥ 345 MPa, tensile strength *R_m_* = 470–630 MPa, elongation *A*_5_ ≥ 22%, impact energy *KV* ≥ 27 J at −20 °C) sheet with the dimensions 500 mm × 500 mm, eight steels with different chromium contents were manually arc surface welded in the horizontal position (PA) with commercial basic-coated electrodes with a diameter of 3.25 mm. The surface welds were single-bead and 150 mm long. Some electrodes were used for the welding of general structural steels (weld no. 1, EN ISO 2560-A: E 42 4 B 32 H5, *R_e_* > 440 MPa, *R_m_* = 510–610 MPa, *A*_5_ > 24%, *KV* > 67 J at −40 °C) and others for welding creep-resisting steels (weld no. 2: EN ISO 3580-A: E Cr Mo 1 B 42 H5, *R_e_* > 470 MPa, *R_m_* = 570–670 MPa, *A*_5_ > 20%, *KV* > 95 J at +20 °C; weld no. 3: EN ISO 3580-A: E Cr Mo 5 B 42 H5, *R_e_* > 490 MPa, *R_m_* = 580–740 MPa, *A*_5_ > 18%, *KV* > 70 J at +20 °C; weld no. 4: EN ISO 3580-A: E Cr Mo 9 B 42 H5, *R_e_* > 550 MPa, *R_m_* = 680–780 MPa, *A*_5_ > 15%, *KV* > 50 J at +20 °C) and for hard surfacing (weld no. 5: EN 14700: E Fe 3, hardness 350–450 HB; welds no. 6 to 8: EN 14700: E Fe 8, hardness 55–60 HRC). A characteristic of all electrodes is alloying from the coating. The chemical compositions of the investigated materials (Si and Mn in S355 steel and weld no.1, while Cr in welds no. 2 to 8) are given in Section 3.1. The welding power source was “Fronius magic waves 3000” and the welding parameters for all welds were the welding current *I* = 100 A ± 1.5 A, the welding voltage *U* = 22.5 V ± 3 V, the welding speed *v_w_* = 2.83 ± 0.35 mm/s, and the arc efficiency *η_a_* = 0.8 [22]. 

For the analysis of the mixing of the weld-pool base and the filler material, the chemical compositions of the filler and base material and of the surface single-bead welds are important. Since the mixing of the weld pool during arc welding is very intensive, it is not possible to determine the chemical composition of the pure filler material from a single-bead surface weld. Therefore, three-layered surface welds were made for the chemical analysis of the pure filler materials, the first layer with five welds, the second layer with four welds, and the third layer with three welds. For the chemical analysis of the pure filler materials, the surfaces of the three-layered surface welds were flattened by milling and finished by grinding them on a grinding machine.

The chemical composition of the base material and the single-bead surface welds and the distribution of the alloying elements through the fusion lines were measured on metallographic cross-sections. Two metallographic cross-sections (¼ and ½ weld lengths from the starting point of the welding) were cut out with a cutting plate from each of the eight welds with the intensive cooling of the samples. All the sections were wet ground with # 80 to # 2400 SiC papers, polished with 1 μm diamond paste, and etched with 3% Nital. This etchant helped to develop the microstructure of the non-alloyed and low-alloyed steels and was therefore ideal for detecting segregations in the different-alloyed steel welds that were welded onto the non-alloy and very low-alloy steel. For the metallographic analysis, no standards were used. Macro cross-sections of single-bead surface welds are shown in Figure 1.

### 2.2. Research Methods

#### 2.2.1. Determining the Chemical Composition 

A JEOL JSM-5610 scanning electron microscope (SEM) and a Gresham Scientific Instruments EDX spectrometer were used to determine the chemical compositions of all the materials. The accelerating voltage used for the EDXS analyses was 20 kV. The chemical compositions of the base and filler materials and the bead-like segregation were determined by a plane-scale analysis. The average chemical compositions (Cr) of the filler materials were determined on the surface of the three-layered surface welds from measurements performed at two different locations (Figure 2a).

The average chemical composition of the base material (Si, Mn) was determined by measuring three different places (Figure 2b) on four metallographic cross-sections—i.e., from 12 measurements.

The average chemical composition of each single-bead surface weld (Si, Mn in weld no. 1, Cr in welds no. 2 to 8) was determined from two metallographic cross-sections. In each cross-section, four measurements were carried out in the part of the filler material and two measurements in the part of the base material (Figure 2b)—i.e., a total of 12 measurements were carried out, from which the average chemical composition of the individual single-bead surface weld was calculated. Prior to the start, a sketch of the measurement site (Figure 2a,b) was given to the SEM operator with a note that the analyzed areas should be as large as possible in accordance with the possibilities of the device in different measured parts of the welds and the base material.

In [1], the average chemical composition of each single-bead surface weld was determined in one metallographic cross-section in the same area, meaning that there were half the number of measurements. The values from [1] were incorporated into the actual study and a comparison of the results for both cases was made. The partially mixed zone near the fusion line was proven by the line analysis to go perpendicularly through the fusion line (Figure 2c).

#### 2.2.2. Determining of the Admixing Rate

The admixing rate of the S355 steel was assessed by comparing the chemical compositions of the filler material and the single-bead surface weld. The admixing rate of the base material into the weld was determined on the basis of the element Cr, because it is only present in the filler material. Accordingly, the final average mixing rate was assessed on seven welds (no. 2 to 8) from a total of 84 measurements (7 welds × 12 measurements). The average admixing rates *D^w^* in each weld were assessed in two ways, which have already been used in [1] but with half the number of measurements:by comparing the Cr content of the filler material (marked *a* next to the number of the surface weld in Table 1) and the average value of all the twelve measured Cr contents (marked *ev* next to the number of the surface weld in Table 1) [1]:
(1)$$D_{ev}^{w}=[1-\left(\frac{{\mathrm{Cr}}^{ev}}{{\mathrm{Cr}}^{a}}\right)]\times 100,\left(\%\right)$$

by comparing the Cr content of the filler material (marked *a* next to the number of the surface weld in Table 1) and the average value of the two smallest measured Cr contents (marked *min* next to the number of the surface weld in Table 1); this is the highest value of the admixing rate of the individual weld [1]:


(2)
$$D_{max}^{w}=[1-\left(\frac{{\mathrm{Cr}}^{min}}{{\mathrm{Cr}}^{a}}\right)]\times 100,\left(\%\right)$$


In the equations, Cr is in wt.%. The final different average admixing rates were calculated as four different arithmetic means:(3)$$D_{ev}=\sum_{2}^{8}D_{ev}^{w}/7,\;(\%)$$
(4)$$D_{max}=\sum_{2}^{8}D_{max}^{w}/7,\;(\%)$$
(5)$$D_{e+m}=\sum_{2}^{8}\left(D_{ev}^{w}+D_{max}^{w}\right)/14,\;(\%)$$
(6)$$D_{sum}^{}=\left(D_{ev}+D_{e+m}\right)/2,\;(\%)$$

The accuracy criterion was the average value *D* = 30%, which was geometrically determined for the same surface welds in [1]. The final values of the admixing rate were controlled by calculating the average values of the Si*^wc^* and Mn*^wc^* contents in surface weld no. 1. Since both elements exist in the filler and the base material, their average content *X^wc^* in the weld is [1]:(7)$$X^{wc}=\left[X_{}^{S355}\times 0.01D\right]+\left[X_{}^{a}\times \left(1-0.01D\right)\right],\;(\mathrm{wt}.\%)$$
where *X*^S^^355^ and *X^a^* are the contents of the element *X* in the base S355 steel and in the filler material (wt.%) and *D* is the admixing rate of the base material (%).

In order to determine the admixing rate with the EDX spectrometer as accurately as possible, it is necessary to know the segregated areas in the welds. Segregations were determined with light microscopy and EDXS analyses in Nital-etched metallographic cross-sections of the surface welds.

## 3. Results and Discussion

### 3.1. Chemical Composition

The chemical compositions of the base S355 steel, the filler materials, and the single-bead surface welds as well as the calculated admixing rates of S355 steel in the individual surface welds are given in Table 1, which also includes the values from [1].

The chemical composition of the single-bead surface welds shows that the weld melt mixed very well during arc welding with the coated electrode. For welding electrodes no. 1 to no. 4, the observed deviation from the average value of the chemical compositions is up to ±0.33 wt.%, and in the case of the electrodes used for surfacing the deviation is greater—up to ±0.74 wt.%. The results show that a greater deviation is expected in more alloyed electrodes. The deviations are therefore the result of the differences between the individual coated electrodes, as well as the result of the manual welding and with this connected oscillation of the welding speed and the arc length during welding. Due to the admixture of the S355 steel in the weld pool, the portions of the alloying elements in single-bead surface welds are 59.9 to 81.7% of their content in the filler material (the average value of the seven welds is 71.2%).

### 3.2. Admixing Rate of the Base Material

The results in Table 1 show that the admixing rates of the S355 steel in individual single-bead surface welds are: from all twelve measurements, $D_{ev}^{w}$ = 18.3–40.1%; from six measurements [1], $D_{ev}^{w*}$ = 20.1–41.2%; from the two lowest Cr contents out of twelve, $D_{max}^{w}$ = 24.6–43.0%; and from the two lowest Cr values out of six [16], $D_{max}^{w*}$ = 20.8–42.9%. All these values are comparable with the literature data for arc welding with coated electrodes [1,2,3,4,10,17]. The final average admixing rates calculated with Equations (3)–(6), and the accuracy of the results with a comparative value *D* = 30% is shown in Table 2.

The results show that of all the values, *D_max_* is the most deviated, always with a positive value having the least accuracy. Equation (6) gives the admixture rate *D_sum_*, which is the closest to the geometrically determined value.

The average value of the admixing rate of the S355 steel was verified and confirmed by calculating the average Si and Mn contents in single-bead surface weld no. 1 using Equation (7). The calculations for the minimum, maximum, and the most competent comparative mixing rate are shown:*D_ev_* = 28.8%: $Si^{wc}=\;\left[0.28\times 0.288\right]\;+\;\left[0.65\times \left(1-0.288\right)\right]\;=0.543$ wt.%;

*D_sum_* = 30%: $Si^{wc}=\;\left[0.28\times 0.30\right]\;+\;\left[0.65\times \left(1-0.30\right)\right]\;=0.539$ wt.%;

*D_max_* = 33.6%: $Si^{wc}=\;\left[0.28\times 0.336\right]\;+\;\left[0.65\times \left(1-0.336\right)\right]\;=0.526$ wt.%;

*D_ev_* = 28.8%: $Mn^{wc}=\;\left[1.07\times 0.228\right]\;+\;\left[0.9\times \left(1-0.228\right)\right]\;=0.939$ wt.%;

*D_sum_* = 30%: $Mn^{wc}=\;\left[1.07\times 0.30\right]\;+\;\left[0.9\times \left(1-0.30\right)\right]\;=0.951$ wt.%;

*D_max_* = 33.6%: $Mn^{wc}=\;\left[1.07\times 0.336\right]\;+\;\left[0.9\times \left(1-0.336\right)\right]\;=0.957$ wt.%.

The results show that all the estimated mixing rates give similar values, which are in the range of the EDXS measurement values.

The admixing rate can also be regularly calculated or verified from the welding parameters and the material constants using the following equation [1,7,8]:


(8)
$$D_{wp}={\left(1+\frac{V_{fm}\cdot E_{bm}}{\eta_{a}\cdot \eta_{m}\cdot U\cdot I-E_{fm}\cdot V_{fm}}\right)}^{-1}\times 100,\;(\%)$$


where *D_wp_* is the calculated admixing rate from the welding parameters, *V_fm_* is the quantity of the melting filler materials (mm^3^/s), *E_fm_* and *E_bm_* are the melting enthalpies of the filler and base materials (J/mm^3^), *η_a_* and *η_m_* are the arc and melting efficiencies, *U* is the welding voltage (V), and *I* is the welding current (A).

By transforming Equation (8), the melting efficiency can be calculated from the known value of the admixing rate and checked with the accuracy of the estimated admixing rate:


(9)
$$\eta_{m}=\frac{\left[\frac{V_{fm}\cdot E_{bm}}{{\left(0.01D_{wp}\right)}^{-1}-1}+\left(E_{fm}\cdot V_{fm}\right)\right]}{\eta_{a}\cdot U\cdot I}$$


The comparable value of the melting efficiency *η_m_* = 0.29 was determined in [1]. If the values *V_fm_* = 36 mm^3^/s [1], *E_fm_* = 10.0 J/mm^3^ [1], *E_bm_* = 10.5 J/mm^3^ [1], *η_a_* = 0.8, *U* = 22.5 V, *I* = 100 A, and *D_wp_* = *D_ev_* = 28.8% or *D_e+m_* = 31.2%, respectively, are inserted into Equation (9), the melting efficiency of *η_m_* = 0.285 or *η_m_* = 0.295 is obtained. Both values match, with a 98% accuracy with the comparable value. With *D_sum_* = 30%, the exact value of the melting efficiency *η_m_* = 0.29 is obtained.

Different results confirm the appropriateness of the determination of the admixing rate based on the chemical composition measurements of the selected areas with the EDX spectrometer. It can be seen that ≥96% of the geometrically determined admixing rate can already be achieved from the average of all the measurements with Equation (3) and a more accurate value can be obtained with Equation (6), which is the closest to the geometrically determined values.

#### Subjective Influence of the SEM Operator on the Final Result

A comparison of the results in Table 2 shows that conducting a larger number of measurements does not necessarily lead to a greater accuracy. The values of the admixing rate calculated with Equations (3)–(5) from a smaller number of chemical composition measurements are closer to the comparative value *D* = 30% than the values calculated from a larger number of measurements. This is for two reasons: one is the presence of segregations that generally exist in arc-welded welds in the vicinity of the fusion line and that generally have a more or less band-line shape; the second is the subjective influence of the SEM operator, which is directly related to the determination of the site and the size of the analyzed area. Typical examples of segregations are shown in Figure 3. The segregation of the S355 steel in a non-alloyed single-bead surface weld is seen as brighter bands that are transversely oriented with respect to the columnar grains of the weld (Figure 3a). The martensitic microstructure of the segregations is clearly separated from the columnar grains, which have the microstructure of acicular ferrite with pro-eutectoid ferrite on its crystal boundaries. In alloyed welds, the segregation of the base S355 steel is clearly visible as the darker lines in the lighter less etched or unetched more alloyed base (Figure 3b,c). Conversely, the more alloyed segregations of the filler material in the less-alloyed weld were seen as bright lines in a darker etched base (Figure 3b). All the segregations were verified using EDXS flat analysis. A narrow, partially mixed layer near the fusion lines was determined by the EDXS line analysis across the fusion line. From the arranged curve of the alloying element, we can determine the width of the partly mixed zone (Figure 3d).

If the SEM operator at the place of the base material segregation analyses too small an area, the estimated admixing rate can be much greater than the real one. Conversely, the admixing rate of the base material will be less than the real one if the SEM operator analyses too small an area with more alloyed filler material segregation or if he/she does not analyze the segregation of the base material at all. Therefore, it is very important that the EDXS analysis of a particular site captures a large enough area of the weld close to the base metal and, at the same time, a sufficiently large area of the well-mixed part of the weld. In order to assess the subjective influence of the SEM operator on the result, a statistical analysis of the size of the measured surfaces of all 84 measurements was carried out. We should mention that, prior to the start of this work, a sketch of the measurement site (Figure 2b) was given to the SEM operator with a note that the analyzed areas should be as large as possible in accordance with the possibilities of the device. Despite the instructions that the analyzed area should be as large as possible, the statistical analysis showed that the surface areas of the measured sites were very different (Figure 4). The prevailing areas were up to 1.5 mm^2^ (61% of all measurements), while only 12% of the measurements were performed on areas of between 2.5 and 3.5 mm^2^. On the other hand, three times as many measurements were made on areas ≤1.0 mm^2^ (36%). The presented statistical analysis shows the possible influence of the SEM operator on the final value of the admixing rate.

However, since there is a high probability that due to the distribution of the segregations of the base material along and close to the fusion line, these areas will not be best covered in the EDX analysis, they must be artificially included in the calculation of the admixing rate. The effect of the base-material segregation is represented by Equation (4) on the basis of the two smallest measured values of Cr in the individual weld, which gives the maximum and, at the same time, excessive admixing rates. The smallest values of the Cr are also taken into account in Equation (3). However, due to the large number of measurements, which also cover a well-mixed area, the final value for the admixing rate is significantly less affected by segregations of the base material.

Equation (4) therefore artificially summarizes the measurement at the site of the segregation of the base material from the insufficiently covered well-mixed melt. The influence of the segregations is taken into account in Equations (5) and (6), with Equation (4) representing the correction factor due to the locally distributed segregations. Therefore, the value obtained by Equation (6) is the closest to the geometrically determined values.

### 3.3. Disadvantages of the EDXS Method in Comparison with the Geometrical Methods

Determining the admixing rate with an EDXS spectrometer is very expensive and time-consuming compared to geometrical methods due to the costly devices needed, the sample preparation, and the mode of operation. Geometrical methods, therefore, have several advantages. The preparation of the cross-sections can be very rough, the etching can be made with macro-etchants (often, the shape of the weld is revealed by heat cutting), and cross-sections are recorded with a camera. The photographs of the cross-sections are expanded in a computer, and with a corresponding program (e.g., Image J) the admixing rate can be determined geometrically from the ratio of the surfaces or the heights. From the height ratio, the admixing rate can be accurately determined from sufficiently enlarged photographs using a ruler (due to the measurement length being in mm, the measurement error decreases with a larger magnification). However, the most important thing is the fact that the geometrical methods are independent of the chemical composition and thus the segregation.

## 4. Conclusions

Single-bead surface welds were arc-welded using eight different coated electrodes with different chromium contents onto a 20-mm-thick plate of S355 steel. Based on the results of the light microscopy and the plane-scan and line-scan EDXS analyses, we can conclude that:The weld pool was very intensely and well mixed during arc surfacing with a coated electrode in the first layer, as evidenced by the small differences in the chemical composition of the different parts of the single-bead welds.Near the fusion line, local band-like non-alloyed segregations of the admixed base material as well as the more-alloyed segregation of the filler material due to alloying from the electrode coating are present.A continuous, narrow segregated layer of a partially mixed zone with a very large chemical gradient exists close to the fusion line, at which the contents of the chemical elements change abruptly from the composition of the base material to the composition of the resulting weld.Using the EDXS method with a sufficiently large number of appropriately selected analyzed areas on the individual cross-section of the single-bead surface weld, the average admixing rate of the base material with an accuracy ≥96% (*D* ≥ 28.8%) of the geometrically determined value (*D* = 30%) could be obtained from the average chemical composition.Four different equations of different arithmetic means were used for the calculation of the admixing rate. The most precise values of the admixing rate were given by Equation (6), which is the arithmetic mean of the two different arithmetic means and in which the influence of the segregations of the base material is taken into account. With this equation, an identical result for the admixing rate to that of the comparative geometric method (*D* = 30%) was obtained.Due to the negative effect of segregations on the result for the admixing rate, the SEM operator must have knowledge of the specific segregated areas that must be incorporated into the measurements. In addition, to obtain a relevant result for the admixing rate, a sufficiently wide range of well-mixed melts must also be incorporated into the measurements.The determination of the admixing rate with an EDXS spectrometer is very expensive and time-consuming compared to using geometrical methods.

## Figures and Tables

**Figure 1 materials-15-00217-f001:**
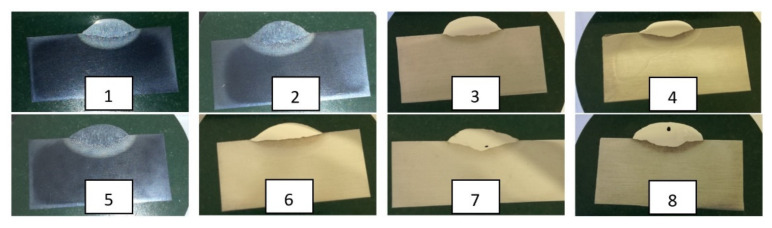
Macro cross-sections of analyzed single-bead surface welds (1, 2, 3, 4, 5, 6, 7, 8 single bead steel welds of eight different alloy-coated electrodes).

**Figure 2 materials-15-00217-f002:**
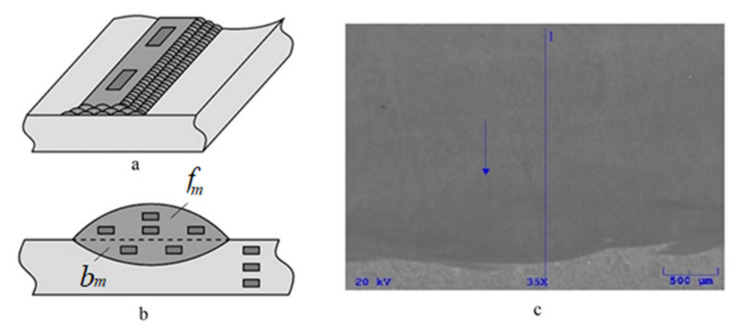
Methods of EDXS analysis: (**a**) Determination of the chemical composition of the filler material-schematically. (**b**) Determination of the chemical composition of the base material and in the single-bead surface welds. Schematically, *f_m_* is the area of the single-bead surface weld belonging to the filler material and *b_m_* is the area belonging to the base material. (**c**) An example of the line analysis through the fusion line (surface weld no. 3).

**Figure 3 materials-15-00217-f003:**
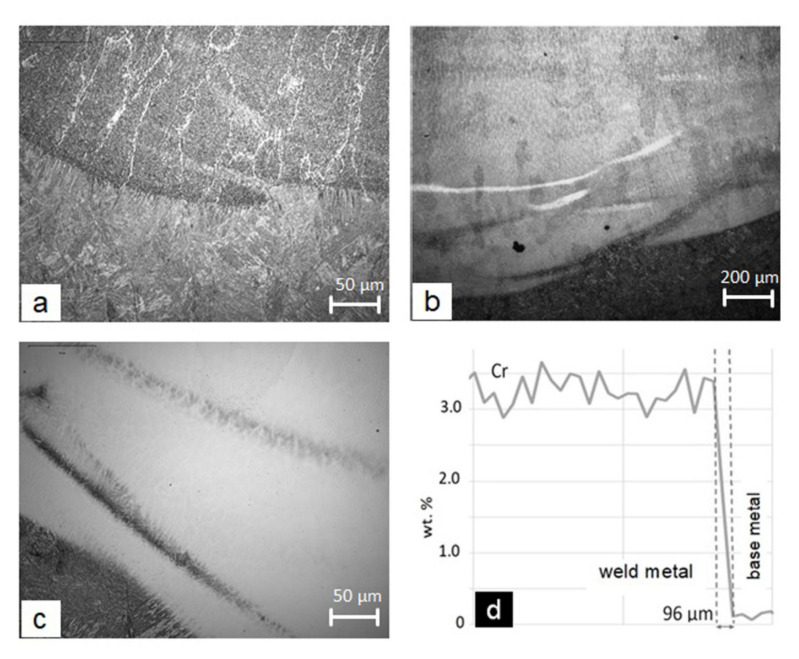
Band-like segregation near the fusion line in surface welds: (**a**) segregations of the S355 steel in the non-alloyed weld no. 1, where columnar crystal grains are also visible (upper half of figure); (**b**) dark segregations of the S355 steel and more alloyed light segregation of the filler materials in the low-alloyed weld no. 2 (the light segregation: Cr = 1.4 wt.%, near the segregation: Cr = 0.66 wt.%); (**c**) dark segregation of the S355 steel in the high-alloyed weld no. 7; (**d**) the distribution of Cr across the fusion line in the weld no. 3, where the partially mixed zone near the fusion line is clearly visible (the width is 96 μm).

**Figure 4 materials-15-00217-f004:**
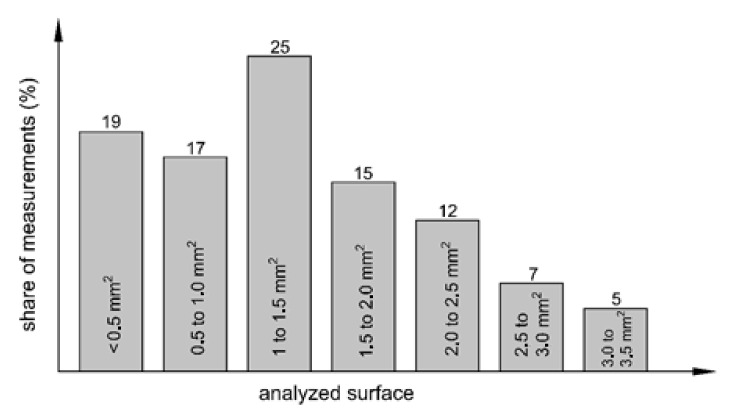
Distribution of the size of the measured surfaces with respect to all 84 measurements.

**Table 1 materials-15-00217-t001:** Contents of the alloying elements Cr, Si, and Mn (wt.%) in the materials and admixing rates *D* (%) of the S355 steel in the individual single-bead surface welds.

Steel	Cr	Cr	Cr	Cr	Cr	Cr	Cr	Cr	$Cr_{ev}^{w}\;$	$Cr_{ev}^{w*}\;$	$\frac{Cr_{ev}^{w}}{Cr}$	$D_{ev}^{w}$	$D_{max}^{w}$	$D_{ev}^{w*}$	$D_{max}^{w*}$
S355	0.28^Si^	1.07^Mn^													
1*^a^*	0.65^Si^	0.90^Mn^													
1*^w^*									0.55^Si^ ± 0.06						
1*^w^*									0.92^Mn^ ± 0.1						
1*^wc^*									0.54^Si^ 0.95^Mn^			30.0			
2*^a^*	1.03 *														
2*^fm^*	0.63 *	0.77 *	0.71 *	0.68 *	0.68	0.75	0.66	0.74	0.70						
2*^bm^*	0.73 *	0.75 *	0.76	0.81					0.76						
2*^ev^*									0.73	0.72 *	0.709	29.1		30.0 *	
2*^min^*	0.63						0.66		0.64	0.65 *	0.621		37.9		36.4 *
3*^a^*	4.55 *														
3*^fm^*	3.24 *	3.33 *	3.17 *	3.28 *	3.31	3.27	3.20	3.38	3.27						
3*^bm^*	3.31 *	3.42 *	3.37	3.46					3.39						
3*^ev^*									3.33	3.31 *	0.732	26.8		27.2 *	
3*^min^*			3.17				3.20		3.18	3.20 *	0.699		30.1		29.7 *
4*^a^*	9.32 *														
4*^fm^*	6.56 *	6.44 *	6.52 *	6.52 *	6.31	6.48	6.58	6.42	6.48						
4*^bm^*	6.63 *	6.47 *	6.86	6.39					6.59						
4*^ev^*									6.53	6.53 *	0.701	29.9		29.9 *	
4*^min^*					6.31			6.42	6.27	6.45 *	0.673		32.7		29.7 *
5*^a^*	1.43 *														
5*^fm^*	0.88 *	0.86 *	0.85 *	0.94 *	0.84	1.04	0.92	0.82	0.89						
5*^bm^*	0.91 *	0.87 *	0.93	1.02					0.93						
5*^ev^*									0.91	0.88 *	0.636	36.4		38.1 *	
5*^min^*			0.85		0.84				0.84	0.88 *	0.587		41.3		40.2 *
6*^a^*	7.10 *														
6*^fm^*	6.14 *	6.17 *	6.26 *	6.34 *	5.67	6.50	5.92	5.45	6.01						
6*^bm^*	5.86 *	5.39 *	5.18	5.98					5.60						
6*^ev^*									5.80	5.93 *	0.817	18.3		16.5 *	
6*^min^*		5.39	5.18						5.28	5.26 *	0.744		25.6		20.8 *
7*^a^*	7.60 *														
7*^fm^*	4.48 *	4.85 *	4.28 *	4.49 *	5.12	4.62	4.63	4.77	4.65						
7*^bm^*	4.42 *	4.39 *	4.51	4.48					4.45						
7*^ev^*									4.55	4.46 *	0.599	40.1		41.2 *	
7*^min^*		4.39	4.28						4.33	4.33 *	0.570		43.0		42.9 *
8*^a^*	7.20 *														
8*^fm^*	5.54 *	5.63 *	5.49 *	5.48 *	5.37	5.53	5.76	4.96	5.47						
8*^bm^*	6.07 *	5.87 *	6.26	5.51					5.93						
8*^ev^*									5.70	5.75 *	0.792	20.8		20.1 *	
8*^min^*			5.49		5.37				5.43	5.49 *	0.754		24.6		23.7 *

*Marks*: 1, 2, 3, 4, 5, 6, 7, 8 single-bead steel welds of eight different alloy-coated electrodes; *a*, measured content in the pure filler material; *fm,* measured content in a single-bead surface weld, part belongs to the filler material; *bm*, measured content in a single-bead surface weld, part belongs to the base material; *ev =* 0.5∙(*fm* + *bm*); *min*, average of the two lowest measured values in a single-bead weld cross-section; ${\mathrm{Cr}}_{ev}^{w}$, average content of Cr in a single-bead surface weld; *w*, contents of Si and Mn in a single-bead surface weld no. 1 from the twelve measurements on the two metallographic cross-sections in the *fm* and *bm* parts; *wc*, average values of Si and Mn in single-bead surface weld no. 1, calculated with the final average admixing rate *D* = 30%; $D_{ev}^{w}$, $D_{max}^{w}$, $D_{ev}^{w*}$, $D_{max}^{w*}$ are the average admixing rates in the individual single-bead surface weld according to Equations (1) and (2); *, values also in [1].

**Table 2 materials-15-00217-t002:** Final average admixing rates *D* and accuracies *A* calculated with different equations.

Equation, *D* (%)/*A* (%)	(3), *D_ev_*/*A*	(4), *D_max_*/*A*	(5), *D_e+m_*/*A*	(6), *D_sum_*/*A*
actual research (84 values)	28.8/96.0	33.6/89.3	31.2/96.1	30.0/100
research in [1] (42 values)	29.0/96.7	32.0/93.7	30.5/98.3	29.75/99.2

## Data Availability

The data presented in this study are available on request from the corresponding author.

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
