# Peer review of "Analysis of the Mixing of Filler and Base Materials in Arc-Welded Single-Bead Surface Welds Using an EDXS Method"

_materials, 2021, doi:10.3390/ma15010217_

Round 1

Reviewer 1 Report

The research examines the mixing and admixing rate of the base S355 steel plate into single-bead surface welds using a plane-scan EDXS analysis on metallographic cross-sections. However, there still exist some problems need to be modified.

  1. Abstract was not clear difficult to understand.
  2. The study showed that a larger number of EDXS measurements do not also mean a more accurate admixing rate. What does it mean?
  3. Segregations together with a sufficiently wide area of the well mixed melt must be analysed for the relevant result – rewrite the statement…difficult to understand
  4. Among them are micro- and macro-segregations, which are the result of a locally poorly mixed welded bath – check the spelling? Do you mean welded path?
  5. Some electrodes are used for the welding of general structural steels (weld no. 1, EN ISO 2560-A: E 42 4 B 32 H5), others for welding creep-resisting steels (weld no. 2: EN ISO 3580-A: E Cr Mo 1 B 42 H5; weld no. 3: EN ISO 3580-A: E Cr Mo 5 B 42 H5; weld no. 4: EN ISO 3580-A: E Cr Mo 9 B 42 H5) and for surfacing (weld no. 5: EN 14700: E Fe 3; welds no. 6 to 8: EN 14700: E Fe 8) – It would be better to provide weld number details in table.
  6. The welding power source was “Fronius magic waves 3000” and the parameters were the welding current I = 100 A ± 1.5 A, the welding voltage U = 22.5 V ± 3 V, the welding speed vw = 2.83 ± 0.35 mm/s, and the arc efficiency ηa = 0.8 – provide more details about process parameters used in this study. On what basis the process parameters were selected? And the type of process used?? And it would be better to present the processing parameters in the form of table.
  7. Nominal Chemical compositions of filler wire and base plate as received condition would be beneficial.
  8. Therefore, three-layered surface welds were made for the chemical analysis of the pure filler materials: the first layer with five welds, the second layer with four welds and the third layer with three welds (Figure 1a). – What is the height, width and depth of the single layer? And also, its overlap between five weld, four welds, and three welds?
  9. Authors were deposited 1st layer with 5 welds 2nd layer with 4 welds and the 3rd layer with 3 welds - The average chemical composition of each single-bead surface weld (Si, Mn in weld no. 1, while Cr in welds no. 2 to 8) was determined from two metallographic cross-sections…What does the weld no.1 and 2 to 8 indicates. Its in 1st layer or 2nd 3rd layer? It would be better to provide how the weld number was indicated and in which layer denotes weld number. 1, 2…etc.
  10. What is the main difference between the previous study [1] and the current study?
  11. What does ev and a indicates? In equation (1) provide notations
  12. How does the EDXS analyses was carried out? How the values are mapped? It would be better to include corresponding light microscopy area where the EDXS analysis was performed?
  13. The chemical compositions of the investigated materials (Si and Mn in S355 steel and weld no.1, while Cr in welds no. 2 to 8) are given in the part 3.1. As mention by authors the section 3.1 indicates …For welding electrodes no.1 to no. 4 the observed deviation from the average value of the chemical compositions is up to ± 0.35 wt.% and in the case of the electrodes for surfacing the deviation is greater, up to ± 0.75 wt.%. Difficult to follow up this statement and what is the average value of chemical composition in weld no.1 and what the authors mentioned about the observed deviation? Authors investigated the chemical compositions of Si and Mn in weld no. 1 and Cr in welds no. 2 to 8…. In the results and discussion authors are discussing about welding electrodes no. 1 to 4…the statements are confusing.
  14. The average value of the admixing rate of the S355 steel was verified and confirmed by calculating the average Si and Mn contents in single-bead surface weld no. 1 using Equation (7). The calculations for the minimum, maximum and for the most competent comparative mixing rate are shown: - why the average value of admixing rate was calculated Si and Mn contents only in weld no.1 why not other welds no. 2 to 8 Cr contents?
  15. The segregation of the S355 steel in a non-alloyed single-bead surface weld is seen as brighter bands that are transversely oriented with respect to the columnar grains of the weld (Figure 2a). Indicate where the columnar grains in Figure 2a provide annotations in the figure.
  16. Table 1. Contents of the alloying elements Cr, Si and Mn (wt. %) in the materials and admixing rates D (%) of the S355 steel in the individual single-bead surface welds – Individual single bead is in 1st layer or 2nd 3rd? In which layer in which weld the individual single bead indicates?
  17. In section 3.2.1 authors mentioned that…However, since there is a high probability that due to the distribution of the segregations of the base material along and close to the fusion line, these areas will not be best covered in the analysis, they must be artificially included in the calculation of the admixing rate. On what basis the admixing rate was calculated by artificially including areas close to the fusion line. It ought to be stated clearly.
  18. The smallest values of the Cr are also taken into account in equation (3), but due to the large number of measurements considered, the final value of the admixing rate is significantly less affected by its. Complete the statement.
  19. There are some grammatical problems with the article which need to be modified. Also, there is no proper link between the subheading and some of the statements are difficult to understand.
  20. What is the main take away from this paper?

Reviewer 2 Report

Dear Authors,

I have read your paper: "Analysis of the mixing of filler and base materials in arc-welded single-bead surface welds using an EDXS method".

It fulfills the aims and scope of Materials journal. Presented investigations are interesting. My suggestions and comments are listed below.

General remarks:

  • You have presented only 22 refeences. Only three of them have been published in last three years. Firstly, you have extended your references. Secondly, you should add more of newly published papers, especially from 2020 and 2021.
  • Please check the style of your paper. In couple of places it is out of template.
  • Please support abstract with quantitative results.
  • I propose to add keywords about used welding process.

Introduction:

  • "[1-19]" - this style of citation is unacceptable. You cannot present so many references in one bracket.
  • S355 steel could be characterized by cold cracking during welding process, which is not mentioned in your paper. You should add this important information into introduction.

Experimental procedure:

  • You have to present chemical composition and mechanical properties (yield point, tensile strength and elongation) of used materials. It will prove, that filler materials were chosen propertly.
  • No information about metallographc tests is presented here. Which stnadard reuirements (no. of standard) were used during examinations? Please mark this number. If you have not used any standard, you should describe why. Moreover, the information about used etchant is missing.
  • Very important factor during pad welding is "dilution". It determines the properties and composition of welded structures, e.g., chemical composition. However, in your paper the dilution is not presented and not calculated in "results". It has to be added and discussed.

Results and Discussion:

  • You should present macrographic photos, which allo to calculate dilution. Without dilution calculation, it is hard to analyse the chemical composition of performed welds. In the literature, it is stated that the higher dilution leads for example to more martensite in the welding joint. Moreover, It has been demonstrated that dilution can significantly affect the metallurgical and mechanical properties of multi-pass welds. The dilution calculations have to be added in thi paper.
  • Presented discussion is quite poor. You have not compared your results to the literature - other papers. You should compar your results with other to show the biggest advantages and underline the novelty of your work.

Conclusions:

  • Please support conclusions with quantitative results.

Round 2

Reviewer 1 Report

No comments

Author Response

We would like to thank you for your time for revision our paper.

Kind regards

Reviewer 2 Report

Dear Authors,

Unfortunatelly, many issues in your paper needs improvements. Your answes are confussing, e.g.,

"

  • Why is this unacceptable? All articles in bracket discuss admixing rate on different ways which is explained later in article with citation."

    Presenting so many references [1-19] in one bracket does not give any wider information. You should discuss these references in more details.
  • "In this study the mechanical properties of filler materials are not important"

    This statement is grotesque. In your paper there is no proof for potential readers, that filler materials were choosen propertly. As I clearly wrote in first round, listing general mechanical properties will underline that materials were proper. It will prove, that research design was appriopriate.

  • We didn´t use any standard for metallographic preparation because we have 30 years of experiences in metallographic examinations on almost all metallic alloys."

    So, it should be clearly stated in the paper, which is not mentioned in this state. Each welding engineer knows, that experience is very important, just like reuq irements from standards.
  • I confirmed my previous comment, that you should present macroscopic photographs in your paper.

Round 3

Reviewer 2 Report

Dear Authors,

Following the Journal requirements: the materials should be describes in details. Following this, I still recommand to show mechanical properties.

Reviewer 

Round 4

Reviewer 2 Report

Dear Authors, 

I really do not understand your behaviour. As I stated from the beggining, yoo should mark the mechanical properties of used Materials. Now, I am writing the review in 4 round of peer review process. Suprising, still I cannot fond relevant information. You have listed the properties of filler materials (congratulations, it was not so taft), but what about properties of base material (S355 steel)?????

I clearly marked, why this information is important, what willl be proved by listing mechanical properties.

I recommand to publish this paper after listing above issue - minor revision . 

I wish you good luck. 
